# TOWARD RELIABLE NEURAL SPECIFICATIONS

## ABSTRACT

Having reliable specifications is an unavoidable challenge in achieving verifiable correctness, robustness, and interpretability of AI systems. Existing specifications for neural networks are in the paradigm of *data as specification*. That is, the local neighborhood centering around a reference input is considered to be correct (or robust). However, our empirical study shows that such a specification is extremely overfitted since usually no data points from the testing set lie in the certified region of the reference input, making them impractical for real-world applications. We propose a new family of specifications called *neural representation as specification*, which uses the intrinsic information of neural networks — neural activation patterns (NAP), rather than input data to specify the correctness and/or robustness of neural network predictions. We present a simple statistical approach to mining dominant neural activation patterns. We analyze NAPs from a statistical point of view and find that a single NAP can cover a large number of training and testing data points whereas ad hoc data-as-specification only covers the given reference data point. To show the effectiveness of discovered NAPs, we formally verify several important properties, such as various types of misclassifications will never happen for a given NAP, and there is no-ambiguity between different NAPs. We show that by using NAP, we can verify the prediction of *the entire input space*, while still recalling 84% of the data. Thus, we argue that using NAPs is a more reliable and extensible specification for neural network verification.

## 1 INTRODUCTION

The advances in deep neural networks (DNNs) have brought a wide societal impact in many domains such as transportation, healthcare, finance, e-commerce, and education. This growing societal-scale impact has also raised some risks and concerns about errors in AI software, their susceptibility to cyber-attacks, and AI system safety (Dieterich & Horvitz, 2015). Therefore, the challenge of verification and validation of AI systems, as well as, achieving trustworthy AI (Wing, 2021), has attracted much attention of the research community. Existing works approach this challenge by building on *formal methods* – a field of computer science and engineering that involves verifying properties of systems using rigorous mathematical specifications and proofs (Wing, 1990). Having a formal specification — a precise, mathematical statement of what AI system is supposed to do is critical for formal verification. Most works (Katz et al., 2017; 2019; Huang et al., 2017; 2020; Wang et al., 2021) use the specification of adversarial robustness for classification tasks that states that the NN correctly classifies an image as a given adversarial label under perturbations with a specific norm (usually $l_\infty$). Generally speaking, existing works use a paradigm of *data as specification* — the robutness of local neighborhoods of reference data points with ground-truth labels is the only specification of correct behaviors. However, from a learning perspective, this would lead to *overfitted* specification, since only local neighborhoods of reference inputs get certified.

As a concrete example, Figure 1 illustrates the fundamental limitation of such overfitted specifications. Specifically, a testing input like the one shown in Fig. 1a can never be verified even if all local neighborhoods of all training images have been certified using the $L_\infty$ norm. This is because adversarial examples like Fig. 1c fall into a much closer region compared to testing inputs (e.g., Fig. 1a), as a result, the truly verifiable region for a given reference input like Fig. 1b can only be smaller. All neural network verification approaches following such data-as-specification paradigm inherit this fundamental limitation regardless of their underlying verification techniques. In order to avoid such a limitation, a new paradigm for specifying what is correct or wrong is necessary.

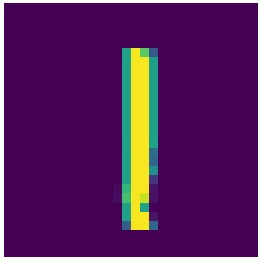 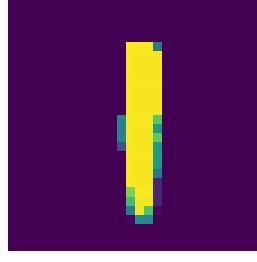 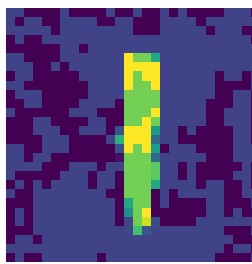 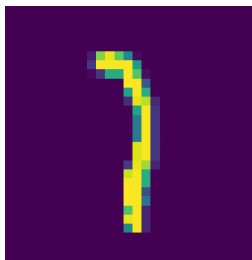

(a) A testing image from MNIST, classified as 1

(b) The closest training image in MNIST, whose $L_\infty$ distance to Fig. 1a is 0.5294

(c) An adversarial example misclassified as 8, whose $L_\infty$ distance to Fig. 1b is 0.2

(d) A testing image, on which our verified NAP (for digit 1) disagrees with the ground truth (i.e., 7)

Figure 1: The limitation of "data-as-specification": First three images show that a test input can be much further away (in $L_\infty$) from its closest train input compared to adversarial examples (the upper bound of a verifiable local region). The last image shows that even data itself can be imperfect.

The intrinsic challenge is that manually giving a proper specification on the input space is no easier than directly programming a solution to the machine learning problem itself. We envision that a promising way to address this challenge is developing specifications directly on top of, instead of being agnostic to, the learned model.

We propose a new family of specifications, *neural representation as specification*, where neural activation patterns form specifications. The key observation is that inputs from the same class often share a *dominant* neural activation pattern (NAP) – a carefully chosen sub-sets of neurons that are expected to be activated (or not activated) for the majority of inputs. Although two inputs are distant in a certain norm in the input space, the neural activations exhibited when the same prediction is made are very close. For instance, we can find a *single* dominant NAP that is shared by *nearly all* training and testing images (including Fig. 1a and Fig. 1b) in the same class but not the adversarial example like Fig. 1c. We can further formally *verify* that *all possible* inputs following this particular dominant NAP can never be misclassified. Specifications based on NAP enable successful verification of a broad region of inputs, which would not be possible if the data-as-specification paradigm were used. For the MNIST dataset, we find that a verifiable dominant NAP *mined* from the training images could cover up to 84% testing images, a significant improvement in contrast to 0% when using neighborhoods of training images as the specification. To our best knowledge, this is the first time that a significant fraction of *unseen* testing images have been formally verified. Interestingly, the verified dominant NAP also enables us to "double check" whether ground truth labels given by human beings are indeed reliable. Fig. 1d shows such an example on which our verified NAP disagrees with the ground truth. We have a tunable parameter to specialize a dominant NAP to avoid accepting such a potentially controversial region of inputs when necessary.

This unique advantage of using NAPs as specification is enabled by the intrinsic information (or neural representation) embedded in the neural network model. Furthermore, such information is a simple byproduct of a prediction and can be collected easily and efficiently. Besides serving as reliable specifications for neural networks, we foresee other important applications of NAPs. For instance, verified NAPs may serve as proofs of correctness or certificates for predictions. We hope our initial findings shared in this paper would inspire new interesting applications. We summarize our contribution as follows:

- We propose a new family of formal specifications for neural networks, *neural representation as specification*, which use activation patterns (NAPs) as specifications.

- We propose a simple yet effective method to mine dominant NAPs from neural networks and training dataset.

- We show that NAPs can be easily checked by out-of-the-box neural network verification tools used in VNNCOMP (2021) – the latest neural network verification competition, such as Marabou.

- We conduct thorough experimental evaluations from both statistical and formal verification perspectives. Particularly, we show that a single dominant NAP is sufficient for certifying a significant fraction of unseen testing images.

## 2 BACKGROUND

### 2.1 NEURAL NETWORKS FOR CLASSIFICATION TASKS

In this paper, we focus on feed-forward neural networks for classification. A feed-forward neural network $\mathcal{N}$ of $L$ layers is a set $\{(\boldsymbol{W}^i, \boldsymbol{b}^i) \mid 1 \leq i \leq L\}$, where $\boldsymbol{W}^i$ and $\boldsymbol{b}^i$ are the weight matrix and the bias for layer $i$, respectively. The neural network $\mathcal{N}$ defined a function $F_N : \mathbb{R}^{d_0} \to \mathbb{R}^{d_L}$ ($d_0$ and $d_L$ represent the input and output dimension, respectively), defined as $F_{\mathcal{N}}(x) = z^L(x)$, where $z^0(x) = x$, $z^i(x) = \boldsymbol{W}^i \sigma(z^{i-1}(x)) + \boldsymbol{b}^i$ and $\sigma$ is the activation function. Neurons are indexed linearly by $v_0, v_1, \cdots$. In this paper we focus only on the ReLU activation function, i.e., $\sigma(x) = \max(x, 0)$ element-wise, but the idea and techniques can be generalized for different activation functions and architectures as well. The $i^{th}$ element of the prediction vector $F_{\mathcal{N}}(x)[i]$ represents the score or likelihood for the $i^{th}$ label, and the one with the highest score ($\arg\max_i F_{\mathcal{N}}(x)[i]$) is often considered as the predicted label of the network $\mathcal{N}$. We denote this output label as $\mathcal{O}_{\mathcal{N}}(x)$. When the context is clear, we omit the subscript $\mathcal{N}$ for simplicity.

### 2.2 ADVERSARIAL ATTACKS AGAINST NEURAL NETWORKS AND THE ROBUSTNESS VERIFICATION PROBLEM

Given a neural network $\mathcal{N}$, the aim of adversarial attacks is to find a perturbation $\delta$ of an input $x$, such that $x$ and $x + \delta$ are "similar" according to some domain knowledge, yet $\mathcal{O}(x) \neq \mathcal{O}(x + \delta)$. In this paper, we use the common formulation of "similarity" in the field: two inputs are similar if the $L_\infty$ norm of $\delta$ is small. Under this formulation, finding an adversarial example can be defined as solving the following optimization problem:[1]

$$\min||\delta||_\infty \text{ s.t } \mathcal{O}(x) \neq \mathcal{O}(x + \delta)$$

In practice, it is very hard to formally define "similar": should an image and a crop of it "similar"? Should two sentences differ by one synonym the same? We refer curious readers to the survey (Xu et al., 2020) for a comprehensive review of different formulations.

One natural defense against adversarial attacks, called *robustness verification*, is to prove that $\min||\delta||_\infty$ must be greater than some user-specified threshold $\epsilon$. Formally, given that $\mathcal{O}(x) = i$, we verify

$$\forall x' \in B(x, \epsilon) \cdot \forall j \neq i \cdot F(x')[i] - F(x')[j] > 0 \tag{1}$$

where $B(x, \epsilon)$ is a $L_\infty$ norm-ball of radius $\epsilon$ centered at $x$: $B(x, \epsilon) = \{x' \mid ||x - x'||_\infty \leq \epsilon\}$. If Eq. (1) holds, we say that $x$ is $\epsilon$-robust.

### 2.3 MARABOU

For this paper, we use Marabou (Katz et al., 2019), a dedicated state-of-the-art NN verifier. Marabou extends the Simplex (Nelder & Mead, 1965) algorithm for solving linear programming with special mechanisms to handle non-linear activation functions. Internally, Marabou encodes both the verification problem and the adversarial attacks as a system of linear constraints (the weighted sum and the properties) and non-linear constraints (the activation functions). Same as Simplex, at each iteration, Marabou tries to fix a variable so that it doesn't violate its constraints. While in Simplex, a violation can only happen due to a variable becoming out-of-bound, in Marabou a violation can also happen when a variable doesn't satisfy its activation constraints.

By focusing only on neural networks with piecewise-linear activation functions, Marabou makes two insights: first, only a small subset of the activation nodes are relevant to the property under consideration. Hence, Marabou treats the non-linear constraints lazily and reduces the number of expensive

---

[1]While there are alternative formulations of adversarial robustness (see Xu et al. (2020)), in this paper, we use adversarial attacks as a black box, thus, stating one formulation is sufficient.

case-splits, making it much faster than traditional SMT solvers. Second, Marabou repeatedly refines each variable's lower and upper bound, hoping that many piecewise-linear constraints can be turned into linear (phase-fixed), reducing further the need for case splitting. Altogether, Marabou achieves state-of-the-art performance on a wide range of benchmarks (VNNCOMP, 2021).

## 3 NEURAL ACTIVATION PATTERNS

In this section, we discuss in detail neural activation patterns (NAP), what we consider as *dominant* NAPs and how to relax them, and what interesting properties of neural activation patterns can be checked using Marabou.

### 3.1 NAPS, DOMINANT NAPS, RELAXED DOMINANT NAPS

In our setting (Section 2), the output of each neuron is passed to the ReLU function before going to neurons of the next layer, i.e., $z^i(x) = \boldsymbol{W}^i \sigma(z^{i-1}(x)) + \boldsymbol{b}^i$. We abstract each neuron into two states: *activated* (if its output is positive) and *deactivated* (if its output is non-positive). Clearly, for any given input, each neuron can be either activated or deactivated, but not both.

**Definition 3.1** (Neural Activation Pattern). A *Neural Activation Pattern (NAP)* of a neural network $\mathcal{N}$ is a pair $\mathcal{P}_{\mathcal{N}} := (A, D)$, where $A$ and $D$ are two disjoint set of neurons.

We say that an input $x$ *follows* a NAP $\mathcal{P}_{\mathcal{N}}$ if after computing $F_{\mathcal{N}}(x)$, the neurons in $A$ are all activated, and neurons in $D$ are all deactivated. We denote this as $\mathcal{P}_{\mathcal{N}}(x) = True$. Note that $A$ and $D$ are not required to contain all activated and deactivated neurons, thus for a given $\mathcal{N}$ an input $x$ can follow multiple NAPs. Our intuition is that for some NAPs, all inputs following them should be classified correctly. We call such NAPs the *dominant* NAPs.

**Definition 3.2** (Dominant NAP with respect to a label $\ell$). Given a label $\ell$ and a neural network $\mathcal{N}$, we define a dominant NAP with respect to $\ell$ – denoted $\mathcal{P}_{\mathcal{N}}^{\ell}$ as

$$\mathcal{P}_{\mathcal{N}}^{\ell} := (A^{\ell}, D^{\ell}) \text{ s.t } \forall x \cdot \mathcal{O}(x) = \ell \iff \mathcal{P}_{\mathcal{N}}^{\ell}(x) = True$$

Note that $x$ is quantified over all the input space, not restricted to a $L_{\infty}$ ball or a train or test set. The dominant NAP is a certificate for the prediction: any input that follows $\mathcal{P}_{\mathcal{N}}^{\ell}(x)$ must be classified as $\ell$ by $\mathcal{N}$. However, dominant NAPs are very hard to find (as hard as verifying the robustness property for the whole input space), or may not exist at all. Thus, we wish to relax the definition of dominant NAPs: First, instead of quantifying $x$ over the input space, we limit $x$ to a dataset. Second, we break the strong $\iff$ condition, keeping only the condition $\mathcal{O}(x) = \ell \implies \mathcal{P}_{\mathcal{N}}^{\ell}(x) = True$. Finally, we relax the implication even further, by allowing a neuron to be added to the NAP even when it is not activated/deactivated in all inputs of the same label. We call this relaxed version the $\delta$-relaxed dominant NAP.

**Definition 3.3** ($\delta$-relaxed dominant NAP). Let $S$ be the dataset, and $S_{\ell}$ be the set of data labeled as $\ell$. Clearly $\forall \ell, S_{\ell} \subseteq S$ and $\bigcup_{\ell} S_{\ell} = S$. We introduce a relaxing factor $\delta \in [0, 1]$. A $\delta$-relaxed dominant NAP for the label $l$ – denoted $\delta.\mathcal{P}_{\mathcal{N}}^{\ell}$ – is defined through construction (mining)

1. Initialize two counters $a_k$ and $d_k$ for each neuron $v_k$.

2. $\forall x \in S_{\ell}$, compute $F_{\mathcal{N}}(x)$, let $a_k \mathrel{+}= 1$ if $v_k$ is activated; let $d_k \mathrel{+}= 1$ if $v_k$ is deactivated.

3. $A^{\ell} \leftarrow \left\{ v_k \mid \frac{a_k}{|S_{\ell}|} \geq 1 - \delta \right\}, D^{\ell} \leftarrow \left\{ v_k \mid \frac{d_k}{|S_{\ell}|} \geq 1 - \delta \right\}$

4. $\delta.\mathcal{P}_{\mathcal{N}}^{\ell} \leftarrow (A^{\ell}, D^{\ell})$

Intuitively, for each neuron, we ask: how often is this neuron activated/deactivated upon seeing inputs from a label $l$. $\delta$ controls how precise we want to be: should we add a neuron to our NAP only if it is activated in all the inputs labeled $\ell$, or 95% of the inputs is enough? Thus, as $\delta$ increases, we expect the size of NAPs also to increase. Choosing the right $\delta$ has a huge impact on how useful the relaxed dominant NAPs can be. Table 1 gives us some intuition why. For MNIST, the table shows how many test images from a label $\ell$ follow $\delta.\mathcal{P}^{\ell}$, together with how many test images from other labels that also follow the same $\delta.\mathcal{P}^{\ell}$. For example, there are 980 images in the test set with

Table 1: The number of the test images in MNIST that follow a given $\delta$.NAP. For a label $i$, $\bar{i}$ represents images with labels other than $i$ yet follow $\delta$.NAP$^i$. The leftmost column is the values of $\delta$. The top row indicates how many images in the test set are of a label.

| | 0 (980) | | 1 (1135) | | 2 (1032) | | 3 (1010) | | 4 (982) | | 5 (892) | | 6 (958) | | 7 (1028) | | 8 (974) | | 9 (1009) | |
|---|---|---|---|---|---|---|---|---|---|---|---|---|---|---|---|---|---|---|---|---|
| | 0 | $\bar{0}$ | 1 | $\bar{1}$ | 2 | $\bar{2}$ | 3 | $\bar{3}$ | 4 | $\bar{4}$ | 5 | $\bar{5}$ | 6 | $\bar{6}$ | 7 | $\bar{7}$ | 8 | $\bar{8}$ | 9 | $\bar{9}$ |
| 0.00 | 967 | 20 | 1124 | 8 | 997 | 22 | 980 | 13 | 959 | 25 | 874 | 32 | 937 | 26 | 1003 | 28 | 941 | 22 | 967 | 12 |
| 0.01 | 775 | 1 | 959 | 0 | 792 | 4 | 787 | 2 | 766 | 3 | 677 | 1 | 726 | 4 | 809 | 2 | 696 | 3 | 828 | 4 |
| 0.05 | 376 | 0 | 456 | 0 | 261 | 1 | 320 | 0 | 259 | 0 | 226 | 0 | 200 | 0 | 357 | 0 | 192 | 0 | 277 | 0 |
| 0.10 | 111 | 0 | 126 | 0 | 43 | 0 | 92 | 0 | 76 | 0 | 24 | 0 | 45 | 0 | 144 | 0 | 44 | 0 | 73 | 0 |

label 0 (second column). Among them, 967 images follow $0.\mathcal{P}^0$. In addition to that, there are 20 images from the other 9 labels that also follow $0.\mathcal{P}^0$. With the increase of $\delta$, we can see that in both cases, both numbers decrease, suggesting that it is harder for an image to follow $0.\mathcal{P}^0$ without being classified as 0 (the NAP is more precise), at the cost of having many images classified as 0 fail to follow $0.\mathcal{P}^0$ (the NAP recalls worse). Essentially, $\delta$ controls the precision-recall trade-off for the NAP. Note that a small increase in $\delta$, such as 0.01, can already have a huge effect: a neuron may be activated in 99% of the input, but the collection of all said neurons is only activated together in 79% of the input (775 out of 980), as can be seen with label 0 in Table 1.

For the rest of the paper, if not explicitly stated, we refer to the $\delta$-relaxed dominant NAPs when talking about NAPs. When the context is clear, we drop any of the $\delta, \mathcal{N}$ and $\ell$ from the notation for clarity.

### 3.2 Neural network properties with Neural Activation Patterns

We expect that NAPs can serve as the key component in more reliable specifications of neural networks. As the first study on this topic, we introduce here two important ones.

**The non-ambiguity property of NAPs** We want our NAPs to give us some confidence about the predicted label of an input, thus a crucial sanity check is to verify that no input can follow two different NAPs of two different labels. Formally, we verify:

$$\forall x \cdot \forall i, j (i \neq j) \cdot \mathcal{P}^i(x) = True \implies \mathcal{P}^j(x) = False \qquad (2)$$

Note that this property is trivial if either $A_{\mathcal{N}}^i \bigcap D_{\mathcal{N}}^j$ or $A_{\mathcal{N}}^j \bigcap D_{\mathcal{N}}^i$ is non-empty: a single input cannot activate and deactivate the same neuron. If that's not the case, we can encode and verify the property using Marabou.

**NAP-augmented robustness property** As seen in Figure 1 and Section 4.1, in many cases the maximum verifiable $B(x, \epsilon)$ does not contain any other input in the dataset. Our key insight is that by adding NAP constraints, we change the problem from verifying a "perfect" but small $L_\infty$ ball to an "imperfect" but much bigger $L_\infty$ ball, which may cover more of the input space. We explain this insight in more detail in Section 3.3. Concretely, we formalize our new robustness verification problem as follows: given a neural network $\mathcal{N}$ and a mined $\mathcal{P}_{\mathcal{N}}^i$, we check

$$\forall x' \in B^+(x, \epsilon, \mathcal{P}_{\mathcal{N}}^i) \cdot \forall j \neq i \cdot F(x')[i] - F(x')[j] > 0$$

in which $\mathcal{O}(x) = i$ and

$$B^+(x, \epsilon, \mathcal{P}_{\mathcal{N}}^i) = \{x' \mid ||x - x'||_\infty \leq \epsilon, \mathcal{P}_{\mathcal{N}}^i(x') = True\}$$

**Working with NAPs using Marabou** NAPs and NAP properties can be encoded using Marabou with little to no changes to Marabou itself. To force a neuron to be activated or deactivated, we add a constraint for its output. To improve performance, we infer ReLU's phases implied by the NAPs, and change the corresponding constraints[2]. For example, given a ReLU $v_i = max(v_k, 0)$, to enforce $v_k$ to be activated, we remove the constraint from Marabou and add two new ones: $v_i = v_k$, and $v_k \geq 0$.

Table 2: The frequency of each ReLU and the dominant NAPs for each label. Activated and deactivated neurons are denoted by $+$ and $-$, respectively, and $*$ denotes an arbitrary neuron state.

| Label | Neuron states | #samples | Dominant NAP |
|---|---|---|---|
| 0 (Green) | $(+,-,-,+,-,+)$ | 8 | $(+,*,-,+,-,+)$ |
|  | $(+,+,-,+,-,+)$ | 2 |  |
| 1 (Red) | $(+,+,-,-,+,-)$ | 7 | $(*,+,-,+,-,*)$ |
|  | $(-,+,-,-,+,-)$ | 2 |  |
|  | $(+,+,-,-,+,+)$ | 1 |  |

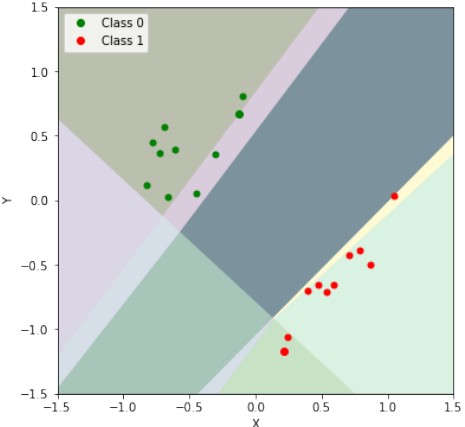

(a) Linear regions in different colors are determined by weights and biases of the neural network. Points colored either red or green constitute the training set.

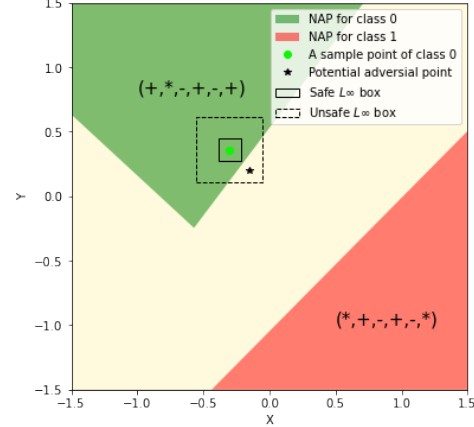

(b) NAPs are more flexible than $L_\infty$ norm-balls (boxes) in terms of covering verifiable regions.

Figure 2: Visualization of linear regions and NAPs as specifications compared to $L_\infty$ norm-balls.

### 3.3 CASE STUDY: VISUALIZING NAPs OF A SIMPLE NEURAL NETWORK

We show the advantages of NAPs as specifications using a simple example of a three-layer feed-forward neural network that predicts a class of 20 points located on a 2D plane. We trained a neural network consisting of six neurons that achieves 100% accuracy in the prediction task. The resulting linear regions as well as the training data are illustrated in Fig. 2a. Table 2 summarizes the frequency of states of each neuron based on the result of passing all input data through the network, and NAPs for labels 0 and 1.

Fig. 2b visualizes NAPs for labels 0 and 1, and the unspecified region which provides no guarantees on data that fall into it. The green dot is so close to the boundary between $\mathcal{P}^0$ and the unspecified region that some $L_\infty$ norm-balls (boxes) such as the one drawn in the dashed line may contain an adversarial example from the unspecified region. Thus, what we could verify ends up being a small box within $\mathcal{P}^0$. However, using $\mathcal{P}^0$ as a specification allows us to verify a much more flexible region than just boxes, as suggested by the NAP-augmented robustness property in Section 3.2. This idea generalizes beyond the simple 2D case, and we will illustrate its effectiveness further with a critical evaluation in Section 4.2.

## 4 EVALUATION

In this section, we validate our observation about the distance between inputs, as well as evaluate our NAPs and NAP properties.

---

[2]Marabou has a similar optimization, but the user cannot control when or if it is applied.

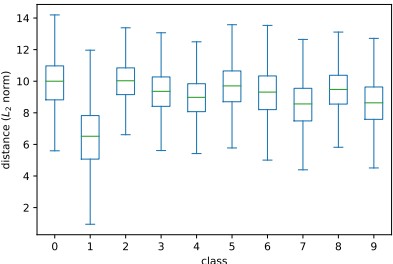 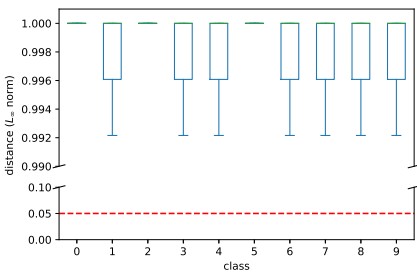

(a) The distribution of $L_2$-norms between any two images from the same label. Images of digit (label) 1 are much similar than that of other digits.

(b) The distribution of $L_\infty$-norms between any two images from the same label. The red line is drawing at 0.05 – the largest $\epsilon$ used in VNNCOMP (2021).

Figure 3: Distances between any two images from the same label (class) are quite significant under different metrics of norm.

### 4.1 TEST DATA ARE FAR AWAY FROM MAXIMUM VERIFIED BOUNDS

We argue that the maximum verifiable bounds are much smaller than the distance between real data, as illustrated in Figure 1. To show that this is not a special case, we plot the distribution of distances in $L_2$ [3] and $L_\infty$ norm between any pair of images with the same label from the MNIST dataset, as shown in Figure 3. We obverse that the smallest $L_\infty$ distance for each label is greater than 0.2, much larger than 0.05, the largest verifiable perturbation $\epsilon$ used in VNNCOMP (2021).

This strongly verifies our motivation that the *data as specification* paradigm is not sufficient to verify data in real-life scenarios. The differences between data of each class are clearly larger than the differences allowed in some certain specifications using $L_\infty$ norm-balls.

### 4.2 THE NAP-AUGMENTED ROBUSTNESS PROPERTY

We conduct two experiments to demonstrate the NAP-augmented robustness property. Both experiments use the MNIST dataset and a 4-layers fully connected network used in VNNCOMP (2021), with 256 neurons for each layer. All checks are run with a 10-minute time limit, on an AMD Ryzen 5565U laptop with 16GBs of RAM. The results are shown in Table 3 and Table 4. For each label $\ell \in [0, 9]$, 'Y' and 'N' indicate that the network is robust for $\ell$ (i.e., no adversarial example) or not, respectively. 'T/o' means the verification of robustness timed out.

The first experiment circles back to Figure 1. In the figure, we show an image $\mathcal{I}$ and its adversarial example that has the distance of 0.2 to $\mathcal{I}$ in $L_\infty$ space. Using our technique, can we prove that there is no adversarial example inside $B^+(\mathcal{I}, 0.2, \delta.\mathcal{P}^1)$? We start with checking if a counter-example exists for any label (first row in Table 3), and somewhat surprisingly, Marabou can only find a counter-example in 3 out of 9 labels within the time limit. Using the most precise NAP $0.\mathcal{P}$, we are able to verify the robustness of $\mathcal{I}$ against 2 labels. By increasing $\delta$ to 0.01, we are able to verify the robustness against *all* other labels. In fact, with $\delta = 0.01$, we are able to verify *the whole input space* (denoted by $\epsilon = 1.0$ in the last row)! We acknowledge a "fineprint" that by using $\delta = 0.01$, we only recall about 84% of the test set (Table 1), but this is still a significant result. To the best of our knowledge, this is the first time a significant region of the input space has been formally verified.

For the second experiment, we verify the all six $(x, \epsilon)$ tuples in MNIST dataset that are known to be not robust from VNNCOMP-2021. We start with augmenting those examples with $0.\mathcal{P}$. Using $0.\mathcal{P}$, we can prove the robustness of one input against all other labels and block adversary examples in many labels for other inputs (Table 4). We also note that adversarial examples still exist against certain labels, indicating that $0.\mathcal{P}$ may be too weak. Indeed, by slightly relaxing the NAP ($\delta = 0.01$), *all* of the chosen inputs can be proven to be robust. Furthermore, with $\delta = 0.01$, we can verify the robustness for 5 of the 6 inputs (Table 4) with $\epsilon = 0.3$ – *an order of magnitude* bigger bound than before. Unfortunately, we are not able to verify bigger $\epsilon$, except for $x_1$ and $x_5$ (that can be verified in

---

[3]The $L_2$ metric is not commonly used by the neural network verification research community as they are less computationally efficient than the $L_\infty$ metric.

Table 3: Verify the robustness of the example in Figure 1

|  | 0 | 1 | 2 | 3 | 4 | 5 | 6 | 7 | 8 | 9 |
|---|---|---|---|---|---|---|---|---|---|---|
| $\epsilon = 0.2$, no NAP | N | - | N | T/o | T/o | T/o | T/o | T/o | N | T/o |
| $\epsilon = 0.2, \delta = 0$ | N | - | N | Y | T/o | T/o | Y | T/o | N | N |
| $\epsilon = 0.2, \delta = 0.01$ | Y | - | Y | Y | Y | Y | Y | Y | Y | Y |
| $\epsilon = 1.0, \delta = 0.01$ | Y | - | Y | Y | Y | Y | Y | Y | Y | Y |

Table 4: Inputs that are not robust can be augmented with a NAP to be robust. We index the 6 known non-robust inputs from MNIST $x_0$ to $x_5$, and augment them with NAP.

|  |  | 0 | 1 | 2 | 3 | 4 | 5 | 6 | 7 | 8 | 9 |
|---|---|---|---|---|---|---|---|---|---|---|---|
| $\mathcal{O}(x_0) = 0$ | $\epsilon = 0.03, \delta = 0$ | - | Y | Y | Y | Y | Y | Y | Y | Y | Y |
|  | $\epsilon = 0.3, \delta = 0.01$ | - | Y | Y | Y | Y | Y | Y | Y | Y | Y |
| $\mathcal{O}(x_1) = 1$ | $\epsilon = 0.05, \delta = 0$ | Y | - | Y | Y | Y | Y | Y | Y | N | Y |
|  | $\epsilon = 0.3, \delta = 0.01$ | Y | - | Y | Y | Y | Y | Y | Y | Y | Y |
| $\mathcal{O}(x_2) = 0$ | $\epsilon = 0.05, \delta = 0$ | - | T/o | T/o | Y | T/o | T/o | Y | N | T/o | T/o |
|  | $\epsilon = 0.3, \delta = 0.01$ | - | Y | Y | Y | Y | Y | Y | Y | Y | Y |
| $\mathcal{O}(x_3) = 7$ | $\epsilon = 0.05, \delta = 0$ | N | T/o | Y | Y | T/o | T/o | Y | - | N | T/o |
|  | $\epsilon = 0.3, \delta = 0.01$ | Y | Y | Y | Y | Y | Y | Y | - | Y | Y |
| $\mathcal{O}(x_4) = 9$ | $\epsilon = 0.05, \delta = 0$ | T/o | Y | Y | Y | Y | Y | N | Y | N | - |
|  | $\epsilon = 0.3, \delta = 0.01$ | Y | T/o | T/o | Y | N | Y | T/o | T/o | T/o | - |
| $\mathcal{O}(x_5) = 1$ | $\epsilon = 0.05, \delta = 0$ | Y | - | N | Y | Y | Y | Y | N | N | N |
|  | $\epsilon = 0.3, \delta = 0.01$ | Y | - | Y | Y | Y | Y | Y | Y | Y | Y |

the whole input space, as demonstrated with the previous experiment). One might tempt to increase $\delta$ even further to verify bigger $\epsilon$, but one must remember that comes with sacrificing the recall of the NAP. Thus, choosing an appropriate $\delta$ is crucial for having useful NAPs.

### 4.3  THE NON-AMBIGUITY PROPERTY OF MINED NAPs

We evaluate the non-ambiguity of our mined NAP at different $\delta$s. At $\delta = 0$, Marabou can construct inputs that follow any pair of NAP, indicating that the 0-relaxed dominant NAPs do not satisfy the property. However, by setting $\delta = 0.01$, we are able to prove the non-ambiguity for *all* pair of NAPs, through both trivial cases and invoking Marabou. Like the previous experiment, we observe the benefit of relaxing $\delta$ here: the most precise NAPs contain so few ReLUs that it is very easy to violate the non-ambiguity property, thus relaxing $\delta$ is crucial for the NAPs to be more useful.

The non-ambiguity property of NAPs holds an important prerequisite for neural networks to achieve a sound classification result. Otherwise, if two labels share the same NAP, the final prediction of inputs with these two labels may also become indistinguishable. We argue that NAPs mined from a well-trained neural network should demonstrate strong non-ambiguity properties and ideally, all inputs with the same label $i$ should follow the same $\mathcal{P}^i$. However, this strong statement may fail even for an accurate model when the training dataset itself is problematic, as what we observed in Fig. 1d as well as many examples in Appendix C. These examples are not only similar to the model but also to humans despite being labeled differently. The experiential results also suggest our mined NAPs do satisfy the strong statement proposed above if excluding these noisy samples.

To conclude, in this section we present a study that verifies our hypothesis about $L$-norm distances between data points in the MNIST dataset, thus explaining the limitation of using $L$-norm in verifying neural networks' robustness. Then, we conduct experiments to show that our $\delta$-relaxed dominant NAPs can be used to verify a much bigger region in the input space, which has the potential to generalize to real-world scenarios.

## 5 RELATED WORK AND FUTURE DIRECTIONS

**Abstract Interpretation in verifying Neural Networks** The software verification problem is undecidable in general (Rice, 1953). Given that a Neural Network can also be considered a program, verifying any non-trivial property of a Neural network is also undecidable. Prior work on neural network verification includes specifications that are linear functions of the output of the network: Abstract Interpretation (AbsInt) (Cousot & Cousot, 1977) pioneered a happy middle ground: by sacrificing completeness, an AbsInt verifier can find proof much quicker, by over-approximating reachable states of the program. Many NN-verifiers have adopted the same technique, such as Deep-Poly (Singh et al., 2019), CROWN (Wang et al., 2021), NNV (Tran et al., 2021), etc. They all share the same insight: the biggest bottle neck in verifying Neural Networks is the non-linear activation functions. By abstracting the activation into linear functions as much as possible, the verification can be many orders of magnitude faster than complete methods such as Marabou. However, there is no free lunch: Abstract-based verifiers are inconclusive and may not be able to verify properties even when they are correct.[4] On the other hand, the *neural representation as specification* paradigm proposed in this work can be naturally viewed as a method of Abstract Interpretation, in which we abstract the state of each neuron to only activated and deactivated by leveraging NAPs. We would like to explore more refined abstractions such as $\{(-\infty], (0, 1], (1, \infty]\}$ in future work.

**Neural Activation Pattern in interpreting Neural Networks** There are many attempts aimed to address the black-box nature of neural networks by highlighting important features in the input, such as Saliency Maps (Simonyan et al., 2014; Selvaraju et al., 2016) and LIME(Ribeiro et al., 2016). But these methods still pose the question of whether the prediction and explanation can be trusted or even verified. Another direction is to consider the internal decision-making process of neural networks such as Neural Activation Patterns (NAP). One popular line of research relating to NAPs is to leverage them in feature visualization (Yosinski et al., 2015; Bäuerle et al., 2022; Erhan et al., 2009), which investigates what kind of input images could activate certain neurons in the model. Those methods also have the ability to visualize the internal working mechanism of the model to help with transparency. This line of methods is known as activation maximization. While being great at explaining the prediction of a given input, activation maximization methods do not provide a specification based on the activation pattern: at best they can establish a correlation between seeing a pattern and observing an output, but not causality. Moreover, moving from reference sample to revealing neural network activation pattern is limiting as the portion of NAP uncovered is dependent on the input data. This means that it might not be able to handle cases of unexpected test data. Conversely, our method starts from the bottom up: from the activation pattern, we uncover what region of input can be verified. This property of our method grants the capability to be generalized. Motivated by our promising results, we would like to generalize our approach to modern deep learning models such as Transformers (Vaswani et al., 2017), which employ much more complex network structures than a simple feed-forward structure.

## 6 CONCLUSION

We propose a new paradigm of neural network specifications, which we call *neural representation as specification*, as opposed to the traditional *data as specifications*. Specifically, we leverage neural network activation patterns (NAPs) to specify the correct behaviours of neural networks. We argue this could address two major drawbacks of "data as specifications". First, NAPs incorporate intrinsic properties of networks which data fails to do. Second, NAPs could cover much larger and more flexible regions compared to $L_\infty$ norm-balls centred around reference points, making them appealing to real-world applications. We also propose a simple method to mine relaxed dominant NAPs and show that working with NAPs can be easily supported by modern neural network verifiers such as Marabou. Through a simple case study and thorough valuation on the MNIST dataset, we show that using NAPs as specifications not only address major drawbacks of *data as specifications*, but also demonstrate important properties such as no-ambiguity and one order of magnitude stronger verifiable bounds. We foresee verified NAPs have the great potential of serving as simple, reliable, and efficient certificates for neural network predictions.

---

[4]Methods such as alpha-beta CROWN (Wang et al., 2021) claim to be complete even when they are Abstract-based because the abstraction can be controlled to be as precise as the original activation function, thus reducing the method back to a complete one.

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

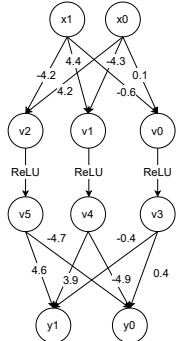

$$v_0 = 0.1x_0 - 0.6x_1$$
$$v_1 = -4.3x_0 + 4.4x_1$$
$$v_2 = 4.2x_0 - 4.2x_1$$
$$v_3 = max(v_0, 0)$$
$$v_4 = max(v_1, 0)$$
$$v_5 = max(v_2, 0)$$
$$y_0 = 0.4v_3 - 4.9v_4 + 3.9v_5 + 6.7$$
$$y_1 = -0.4v_3 + 3.9v_4 + 4.6v_5 - 7.4$$
$$x_0 \leq 0.1 \wedge x_0 \geq 0.02$$
$$x_1 \leq 0.1 \wedge x_1 \geq 0.02$$
$$0 < y_0 - y_1$$

$$v_0 = 0.1x_0 - 0.6x_1$$
$$v_1 = -4.3x_0 + 4.4x_1$$
$$v_2 = 4.2x_0 - 4.2x_1$$
$$v_3 = v_0$$
$$v_4 = max(v_1, 0)$$
$$v_5 = 0$$
$$y_0 = 0.4v_4 - 4.9v_5 + 3.9v_6 + 6.7$$
$$y_1 = -0.4v_4 + 3.9v_5 + 4.6v_6 - 7.4$$
$$x_0 \leq 0.3 \wedge x_0 \geq 0$$
$$x_1 \leq 0.3 \wedge x_1 \geq 0$$
$$v_0 \geq 0$$
$$v_2 \leq 0$$

(a) XNET: A NN that computes the analog XOR function.

(b) Marabou's system of constraints for verifying that XNET is 0.04-robust at (0.06, 0.06)

(c) Check if $\mathcal{P}^1 = ((z_0), ())$ and $\mathcal{P}^0 = ((), (z_2))$ are non-ambiguous in the first quadrant using Marabou

Figure 4: Using Marabou to verify NAP properties of XNET.

## A  A RUNNING EXAMPLE

To help with illustrating later ideas, we present a two-layer feed-forward neural network XNET (Figure 4a) to approximate an analog XOR function $f(x_0, x_1) : [[0, 0.3] \cup [0.7, 1]]^2 \rightarrow \{0, 1\}$ such that $f(x_0, x_1) = 1$ iff $(x_0 \leq 0.3 \wedge x_1 \geq 0.7)$ or $(x_0 \geq 0.7 \wedge x_1 \leq 0.3)$. The network computes the function

$$F_{\text{XNET}}(x) = \boldsymbol{W}^1 \max(\boldsymbol{W}^0(x) + \boldsymbol{b}^0, 0) + \boldsymbol{b}^1$$

where $x = [x_0, x_1]$, and values of $\boldsymbol{W}^0, \boldsymbol{W}^1, \boldsymbol{b}^0, \boldsymbol{b}^1$ are shown in edges of Figure 4a. $\mathcal{O}(x) = 0$ if $F_{\text{XNET}}(x)[0] > F_{\text{XNET}}(x)[1]$, $\mathcal{O}(x) = 1$ otherwise.

Note that the network is not arbitrary. We have obtained it by constructing two sets of 1 000 randomly generated inputs, and training on one and validating o the other until the NN achieved a perfect F1-score of 1.

## B  OTHER EVALUATIONS

### B.1  $L_1$-NORMS OF DISTANCE

Figure 5 shows the distributions of $L_1$-norms of all image pairs from the same class, similar to Fig. 3, the distances between image pairs from class 1 are much smaller compare to other classes.

### B.2  OVERLAP RATIO

Table 5: The maximum overlap ratio for each label (class) on a given $\delta$.NAP for MNIST. Each cell is obtained by $\max_i |N_{col}^\delta \bigcap N_i^\delta| / |N_{col}^\delta|$ where $N_{col}^\delta$ is the set of neurons in the dominant pattern for the label (class) in the header of the column of the selected cell with the given $\delta$, $N_i$ is the set of neurons in the dominant pattern for the label (class) $i$ with the given $\delta$.

|  | 0 | 1 | 2 | 3 | 4 | 5 | 6 | 7 | 8 | 9 |
|---|---|---|---|---|---|---|---|---|---|---|
| 0.00 | 0.959 | 0.928 | 0.963 | 0.966 | 0.972 | 0.973 | 0.930 | 0.965 | 0.957 | 0.981 |
| 0.01 | 0.844 | 0.834 | 0.911 | 0.901 | 0.881 | 0.898 | 0.895 | 0.884 | 0.880 | 0.908 |
| 0.05 | 0.864 | 0.885 | 0.909 | 0.904 | 0.915 | 0.908 | 0.899 | 0.897 | 0.890 | 0.893 |
| 0.10 | 0.877 | 0.900 | 0.910 | 0.901 | 0.921 | 0.910 | 0.890 | 0.899 | 0.900 | 0.901 |
| 0.15 | 0.876 | 0.904 | 0.904 | 0.900 | 0.919 | 0.913 | 0.893 | 0.907 | 0.904 | 0.900 |
| 0.25 | 0.893 | 0.922 | 0.913 | 0.912 | 0.928 | 0.925 | 0.905 | 0.916 | 0.916 | 0.913 |
| 0.50 | 0.903 | 0.905 | 0.925 | 0.923 | 0.926 | 0.923 | 0.907 | 0.918 | 0.927 | 0.927 |

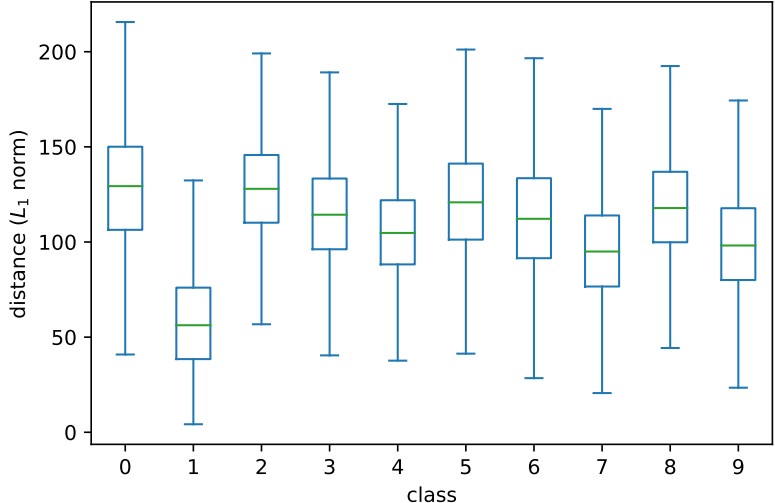

Figure 5: The distribution of $L_1$-norms of all image pairs for each class.

Figure 6 shows the heatmap of the overlap ratio between any two classes for 6 $\delta$ values. For the grid in each column in a heatmap, the overlap ratio is calculated by the number of overlapping neurons of the NAPs of the class labelled for the row and the column divided by the number of neurons in the NAP of the class labelled for the column, which is why the values in the heatmaps are not symmetric along the diagonal. Based on the shade of the colors in our heapmap, we can see that, during the process of increasing $\delta$, the overlapping ratios decrease first and then increase in general, it might because that, with the loose of restriction on when a neuron is considered as activated/inactivated, more neurons are included in the NAP, which means more constrains, but at the same time, for two NAPs of any two classes, it is more likely that they have more neurons appearing in both NAPs.

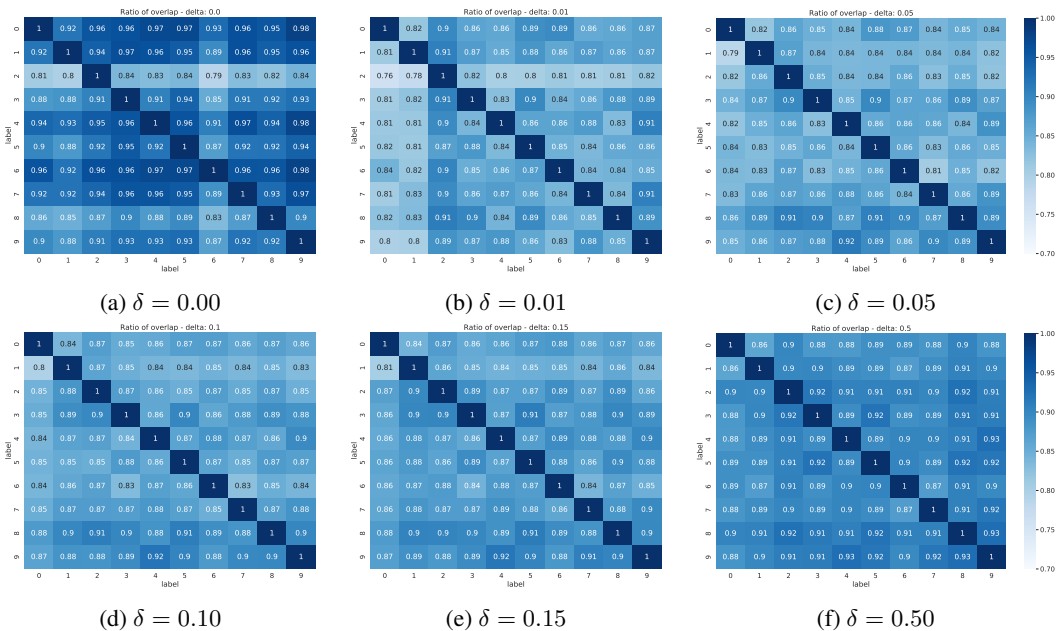

Figure 6: Overlap ratio of the dominant pattern of two labels (classes) on a given $\delta$.NAP. Values in each grid are obtained by $|N_{col}^{\delta} \bigcap N_{row}^{\delta}|/|N_{col}^{\delta}|$ where $N_{col}^{\delta}$ is the set of neurons in the dominant pattern for the label (class) of the column of the selected grid with the given $\delta$, $N_{row}$ is the set of neurons in the dominant pattern for the label (class) of the row of the selected grid with the given $\delta$.

Table 5 shows the maximum overlap ratio for one class, that is, for one reference class, the maximum overlap ratio between this reference class and any other class. This table is basically extracting the maximum values of each column other than the 1 on the diagonal in our heatmap in Fig. 6, in the each column of our table, it also follows the pattern that the value of overlap ratio decreases first and then increase with the increase of $\delta$.

## C MISCLASSIFICATION EXAMPLES

In this section, we display some interesting exmaples from the the MNIST test set that follow the NAP of some class other than their ground truth, which means these images are misclassified. We consider these samples interesting because, instead of misclassification, it is more reasonable to say that these images are given wrong ground truth from human perspective.

delta=0.0, pred=0, gt=5     delta=0.0, pred=0, gt=8     delta=0.0, pred=0, gt=9

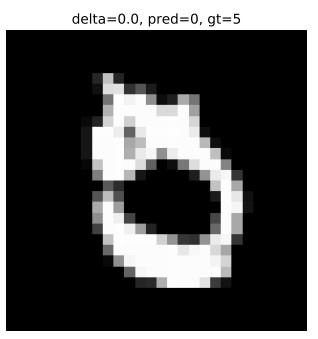 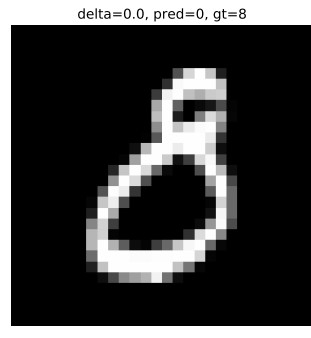 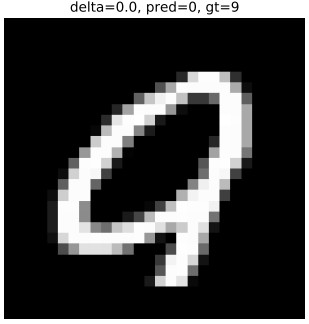

(a) A testing image from MNIST with ground truth 5, classified as 0

(b) A testing image from MNIST with ground truth 8, classified as 0

(c) A testing image from MNIST with ground truth 9, classified as 0

Figure 7: Some interesting test images from MNIST that are misclassified as 0 and also follow the NAP of class 0.

delta=0.0, pred=1, gt=6     delta=0.0, pred=1, gt=7     delta=0.0, pred=1, gt=9

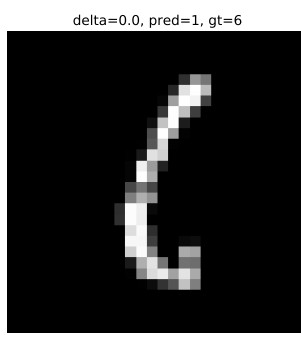 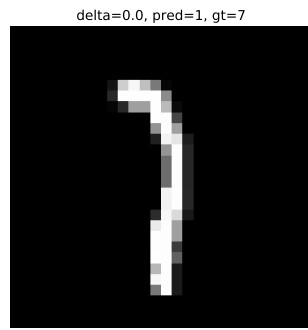 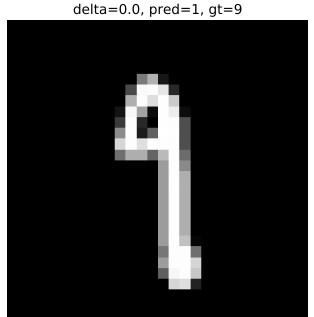

(a) A testing image from MNIST with ground truth 6, classified as 1

(b) A testing image from MNIST with ground truth 7, classified as 1

(c) A testing image from MNIST with ground truth 9, classified as 1

Figure 8: Some interesting test images from MNIST that are misclassified as 1 and also follow the NAP of class 1.

delta=0.0, pred=2, gt=1       delta=0.0, pred=2, gt=3       delta=0.0, pred=2, gt=7

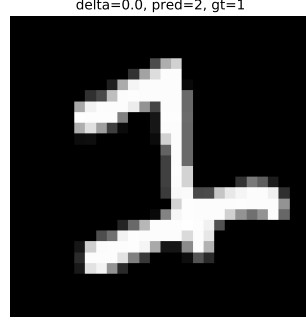 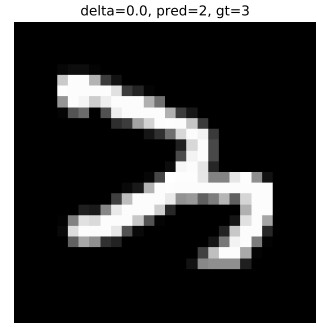 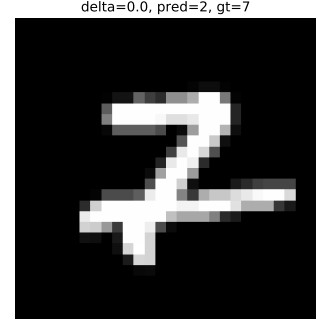

(a) A testing image from MNIST with ground truth 1, classified as 2

(b) A testing image from MNIST with ground truth 3, classified as 2

(c) A testing image from MNIST with ground truth 7, classified as 2

Figure 9: Some interesting test images from MNIST that are misclassified as 2 and also follow the NAP of class 2.

delta=0.0, pred=3, gt=5       delta=0.0, pred=3, gt=7       delta=0.0, pred=3, gt=9

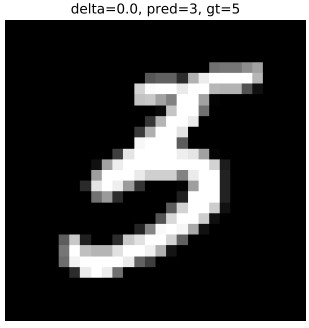 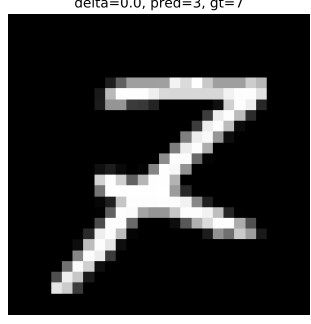 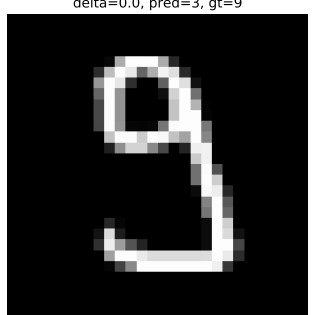

(a) A testing image from MNIST with ground truth 5, classified as 3

(b) A testing image from MNIST with ground truth 7, classified as 3

(c) A testing image from MNIST with ground truth 9, classified as 3

Figure 10: Some interesting test images from MNIST that are misclassified as 3 and also follow the NAP of class 3.

delta=0.0, pred=4, gt=2       delta=0.0, pred=4, gt=6       delta=0.0, pred=4, gt=9

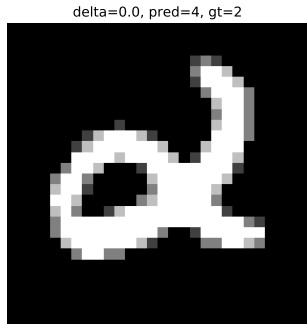 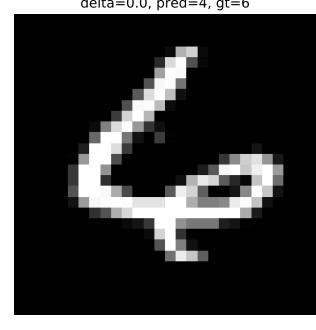 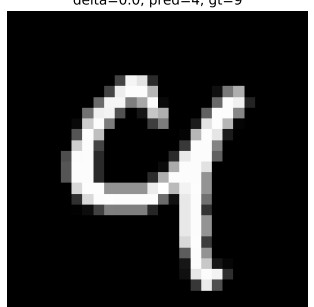

(a) A testing image from MNIST with ground truth 2, classified as 4

(b) A testing image from MNIST with ground truth 6, classified as 4

(c) A testing image from MNIST with ground truth 9, classified as 4

Figure 11: Some interesting test images from MNIST that are misclassified as 4 and also follow the NAP of class 4.

delta=0.0, pred=5, gt=3    delta=0.0, pred=5, gt=6    delta=0.0, pred=5, gt=8

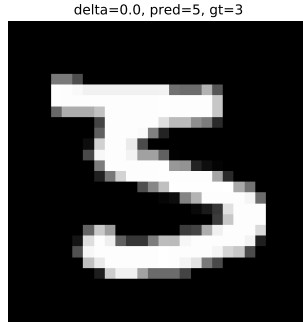 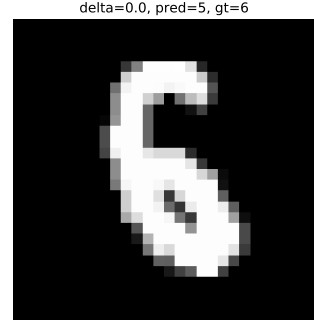 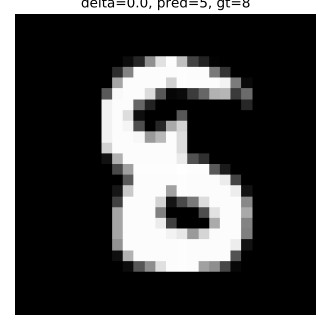

(a) A testing image from MNIST with ground truth 3, classified as 5

(b) A testing image from MNIST with ground truth 6, classified as 5

(c) A testing image from MNIST with ground truth 8, classified as 5

Figure 12: Some interesting test images from MNIST that are misclassified as 5 and also follow the NAP of class 5.

delta=0.0, pred=6, gt=0    delta=0.0, pred=6, gt=2    delta=0.0, pred=6, gt=5

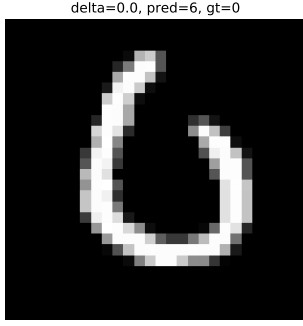 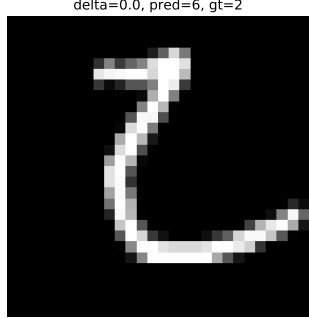 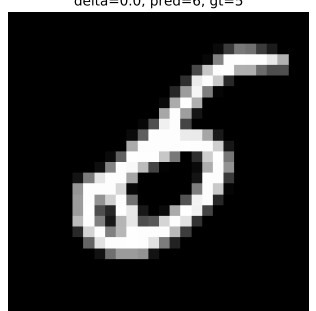

(a) A testing image from MNIST with ground truth 0, classified as 6

(b) A testing image from MNIST with ground truth 3, classified as 6

(c) A testing image from MNIST with ground truth 5, classified as 6

Figure 13: Some interesting test images from MNIST that are misclassified as 6 and also follow the NAP of class 6.

delta=0.0, pred=7, gt=2    delta=0.0, pred=7, gt=3    delta=0.0, pred=7, gt=9

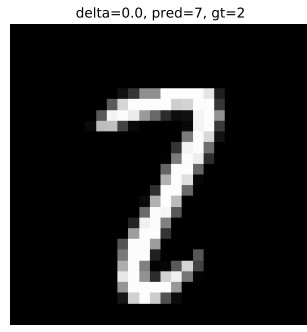 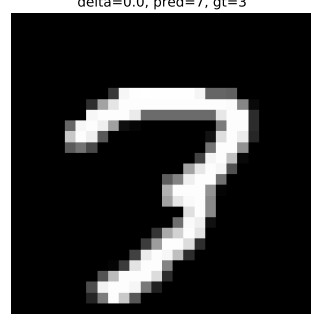 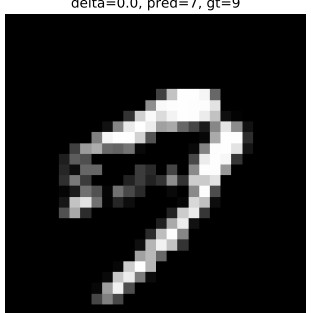

(a) A testing image from MNIST with ground truth 2, classified as 7

(b) A testing image from MNIST with ground truth 3, classified as 7

(c) A testing image from MNIST with ground truth 9, classified as 7

Figure 14: Some interesting test images from MNIST that are misclassified as 7 and also follow the NAP of class 7.

delta=0.0, pred=8, gt=1

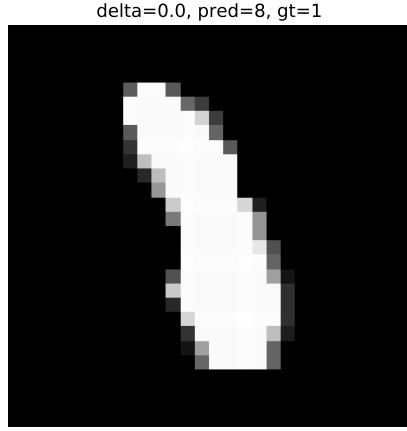

(a) A testing image from MNIST with ground truth 1, classified as 8

delta=0.0, pred=8, gt=3

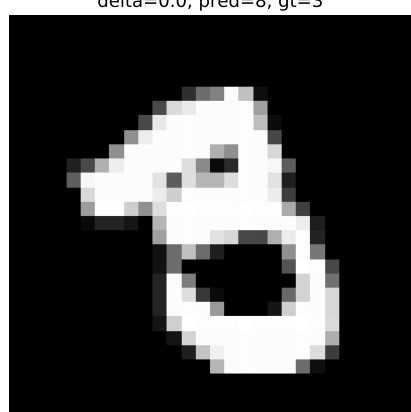

(b) A testing image from MNIST with ground truth 3, classified as 8

Figure 15: Some interesting test images from MNIST that are misclassified as 8 and also follow the NAP of class 8.

delta=0.0, pred=9, gt=4

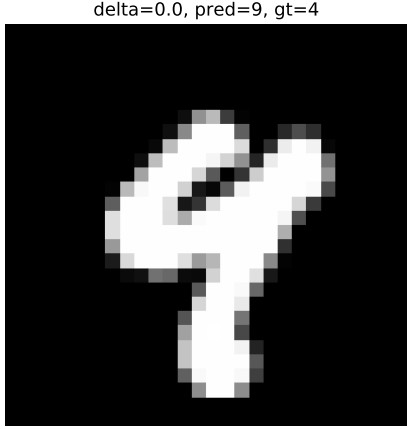

(a) A testing image from MNIST with ground truth 4, classified as 9

delta=0.0, pred=9, gt=7

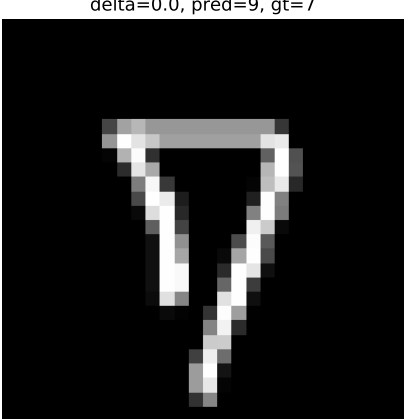

(b) A testing image from MNIST with ground truth 7, classified as 9

Figure 16: Some interesting test images from MNIST that are misclassified as 9 and also follow the NAP of class 9.

