# OpenReview forum: "TOWARD RELIABLE NEURAL SPECIFICATIONS"
_ICLR.cc/2023/Conference — Submitted to ICLR 2023_

### Official Review · Reviewer_rBXu · 2022-10-23

**Confidence:** 3
**Correctness:** 3
**Technical Novelty And Significance:** 3
**Empirical Novelty And Significance:** 2
**Recommendation:** 3

**Clarity, Quality, Novelty And Reproducibility:**

Overall the paper is clear. The parts and motivation about why dominance may matter is neural networks could be explained a bit better (e.g., with examples) as it seems to be the main premise of the NAP specifications.

**Strength And Weaknesses:**

# Strengths:
- The paper highlights an important issue in adversarial robustness certification (the adversarial robustness verification only covers a minuscule part of the entire input space).
- The paper introduces a novel set of specifications and shows that it covers a larger part of the test inputs.

# Weaknesses:
The main issue with the submission is the lack of proper evaluation of the proposed specifications. Particularly, the paper validates the specification on a 2D example and MNIST. The MNIST dataset is severely outdated and known to have artifacts that are not present in other datasets, such as prototypical samples or high certifiable accuracy (compared to, for instance, CIFAR where the top accuracy and the top certified accuracy have a huge gap).
Additionally, the paper only assesses MLP networks.
Consequently, the paper provides insufficient experimental validation of whether the proposed specifications are actually useful and reasonable in practice. It is vital to check if the hypothesized NAP specification works on larger and more diverse datasets, different types of data, and different structures of networks

Moreover, neural activation patterns have already been used in the verification literature of neural networks [1], making the concept less novel.


[1] Henzinger et al. Outside the Box: Abstraction-Based Monitoring of Neural Networks. 2020

**Summary Of The Paper:**

The paper proposed neural activation patterns as a replacement for adversarial robustness specifications to reason about the robustness of the network. The paper argues that this resolves the problem that test samples don't fall into certified regions of an adversarially robust verified network.

**Summary Of The Review:**

Interesting concept of overcoming the problem of certified adversarial regions having little relevance for the test data. Insufficient experimental validation of whether the proposed specifications are actually reasonable in practice.

---

> ### Author Response · Authors · 2022-11-12
> **Discussion about related works. Results for CIFAR10 and CNNs are added.**
>
> > The main issue with the submission is the lack of proper evaluation of the proposed specifications.
>
> This is a great suggestion. We have extended our experiments to a bigger dataset (CIFAR10) and a more complicated network structure ( CNN).
>
> Results for CIFAR10 (the biggest image dataset in VNNCOMP2021) with a convolution net used in the same competition. We extract all (input, epsilon) pairs that are known to be not robust from the competition and index them x_0 to x_6. Without our specification (i.e. augmented NAP), it is impossible to verify an eps larger than 0.012
> | **Epsilon**    | 0.012 |      |     | 0.024 |      |     | 0.12 |      |     |
> |--------------|-------|------|-----|-------|------|-----|------|------|-----|
> | **Delta**  | 0.01  | 0.05 | 0.1 | 0.01  | 0.05 | 0.1 | 0.01 | 0.05 | 0.1 |
> | O(x_0)=8 | **Y**    | **Y**   | **Y**  | N     | T/o  | **Y**  | T/o  | **Y**   | **Y**  |
> | O(x_1)=6 | T/o   | N    | **Y**  | N     | N    | **Y**  | N    | N    | **Y**  |
> | O(x_2)=0 | **Y**    | **Y**   | **Y**  | **Y**    | **Y**   | **Y**  | N    | N    | N   |
> | O(x_3)=1 | N     | N    | N   | N     | N    | N   | N    | N    | N   |
> | O(x_4)=9 | N     | **Y**   | **Y**  | N     | N    | N   | N    | N    | N   |
> | O(x_5)=7 | **Y**   | **Y**   | **Y**  | N     | T/o  | **Y**  | N    | **Y**   | Crash  |
> | O(x_6)=3 | **Y**    | **Y**   | **Y**  | **Y**    | **Y**   | **Y**  | N    | N    | N   |
>
>
> > Moreover, neural activation patterns have already been used in the verification literature of neural networks [1], making the concept less novel.
> [1] Henzinger et al. Outside the Box: Abstraction-Based Monitoring of Neural Networks. 2020
>
> Thanks for mentioning this work. Our work differs from Henzinger et al in two significant aspects:
> First, Henzinger et al does not do verification, but considers the problem of runtime novelty detection through monitoring the internal state of the neural network.  Second, we abstract the internal state differently: Henzinger et al uses box/interval abstraction while we use (relaxed) activation patterns.

---

### Official Review · Reviewer_epj3 · 2022-10-25

**Confidence:** 3
**Correctness:** 3
**Technical Novelty And Significance:** 2
**Empirical Novelty And Significance:** 2
**Recommendation:** 5

**Clarity, Quality, Novelty And Reproducibility:**

The paper is well written and presented. The contribution is not particularly
novel as the main contribution restricts specifications for neural network
verification, thus providing a small increment to previous work. Still, the
specifications proposed are significant to check as in conjunction with
standard specifications for small perturbation radiuses can give stronger
robustness guarantees. The significance of this is  however often
overemphasised by what I think are unsound claims pertaining to the
impracticality of existing specifications.

**Strength And Weaknesses:**

+ The specifications introduced present in my opinion a step towards obtaining
stronger neural network robustness guarantees when used in conjunction with
existing specifications.

- Unsound claim that standard specifications are impractical for real world
  applications. I think that checking small perturbations radiuses which do not
  alter the semantics of the input is important for any application.

- Not surprising empirical results. As the specifications proposed are
  restricted to certain activation patterns, it is expected that the
  perturbation radiuses that can be verified will be bigger when compared to
  standard specifications.

- The evaluation is limited in that it only concerns one benchmark.

**Summary Of The Paper:**

The paper introduces a novel specification for neural network verification,
where instead of checking whether sets of inputs are classified equivalently by
a network, it concerns checking the robustness of networks when their behaviour
is restricted to certain neural activation patterns.

**Summary Of The Review:**

Whilst the paper introduces novel neural network specifications that are
important to check in practice, the overall novelty and significance of the
empirical evaluation are weak.

---

> ### Author Response · Authors · 2022-11-12
> **Clarifications for questions about other works and expectation about the results. Results for CIFAR10 with ConvNet are added.**
>
> Thank you for the careful review. We hope to clarify some of the points brought up in our responses below. This review points to several edits which we will certainly include in a revision.
> >  While the proposed specification present a step towards obtaining stronger neural network robustness guarantees, existing specifications are still important and should not be belittled.
>
> We apologize, this was not our intention. We argue for additional specifications, not against existing ones! We will revise our text to better reflect our intention.
> > The results are somewhat expected: the perturbation radiuses that can be verified will be bigger when compared to standard specifications due to added constraints.
>
> We agree that the verifiable bounds are expected to be bigger when we restrict the activation patterns. However, it was surprising to us how big of a perturbation (10x increase in epsilon, which translates to 10^(number of input dimensions) in the input space) we can cover given a relatively small number of fixed ReLUs (less than 10%), and that the results are not restricted to just MNIST with fully connected NNs,  but also CIFAR10 with CNNs. (see below)
> > The evaluation is limited in that it only concerns one benchmark.
>
> We thank the reviewer for the valuable suggestion. We have extended our experiments to a bigger dataset (CIFAR10) and a more complicated network structure ( CNN).
>
> Results for CIFAR10 (the biggest image dataset in VNNCOMP2021) with a convolution net used in the same competition. We extract all (input, epsilon) pairs that are known to be not robust from the competition and index them x_0 to x_6. Without our specification (i.e. augmented NAP), it is impossible to verify an eps larger than 0.012
> | **Epsilon**    | 0.012 |      |     | 0.024 |      |     | 0.12 |      |     |
> |--------------|-------|------|-----|-------|------|-----|------|------|-----|
> | **Delta**  | 0.01  | 0.05 | 0.1 | 0.01  | 0.05 | 0.1 | 0.01 | 0.05 | 0.1 |
> | O(x_0)=8 | **Y**    | **Y**   | **Y**  | N     | T/o  | **Y**  | T/o  | **Y**   | **Y**  |
> | O(x_1)=6 | T/o   | N    | **Y**  | N     | N    | **Y**  | N    | N    | **Y**  |
> | O(x_2)=0 | **Y**    | **Y**   | **Y**  | **Y**    | **Y**   | **Y**  | N    | N    | N   |
> | O(x_3)=1 | N     | N    | N   | N     | N    | N   | N    | N    | N   |
> | O(x_4)=9 | N     | **Y**   | **Y**  | N     | N    | N   | N    | N    | N   |
> | O(x_5)=7 | **Y**   | **Y**   | **Y**  | N     | T/o  | **Y**  | N    | **Y**   | Crash  |
> | O(x_6)=3 | **Y**    | **Y**   | **Y**  | **Y**    | **Y**   | **Y**  | N    | N    | N   |

---

### Official Review · Reviewer_uPcc · 2022-10-27

**Confidence:** 5
**Correctness:** 4
**Technical Novelty And Significance:** 3
**Empirical Novelty And Significance:** 3
**Recommendation:** 8

**Clarity, Quality, Novelty And Reproducibility:**

The paper is well-written and very clear.
The contribution is original to my knowledge. Could the author discuss the relation of their work to [a,b]?

[a] Gopinath et al, "Property Inference for Deep Neural Networks."

[b] Sabour et al, "Adversarial Manipulation of Deep Representations."

**Strength And Weaknesses:**

Strength:

I really like this work. Adding activation patterns to the pre-conditions of the specification is a simple but clever extension. It seems to be able to cover more test images than the canonical way does, where the constraint is only put on the input. Instead of only having the activation patterns as the pre-condition, still specifying a (now larger) epsilon ball in the input region not only makes the specification more relevant (as it focus the verification on valid images of digits instead of some noise input) but also likely to make the verification task easier.
This work might also motivation development of verification techniques that can handle constraints on NAP more effectively.

Weakness:

The experiment is performed only on small neural networks. The verification is in general challenge, and using tools other than Marabou is unlikely to suddenly allow us to scale to SoTA perception networks. However, I would still be curious to see how scalable the NAP extraction procedure is and what the neural specifications look like on more complex image dataset and networks.


**Summary Of The Paper:**

This paper proposes a novel robustness specification for neural networks. Unlike canonical specifications where pre-conditions are defined as constraints on the neural network input,  the specification used in this paper are defined as particular activation patterns. The paper presents a strategy to extract activation patterns, shows how the specification can be checked with existing verifiers, and how this specification achieves better coverage of the test set.

**Summary Of The Review:**

Overall, I think the paper contains several good and novel ideas and could motivate future work both in application and verification techniques.

---

> ### Author Response · Authors · 2022-11-12
> **Discussion about related works**
>
> We’d like to thank you sincerely for your thoughtful review and we are grateful that you find our work as exciting as we do! We thank the reviewer for sharing these two very related works, which we did not realize but would love to incorporate into our related work discussion.
>
> [a]
> Gopinath et al mainly focus on neural network explanation by studying input properties and layer properties. The latter property is the most related to our work.  Gopinath et al collect activations of all neurons in a specific layer and then learn a decision tree by viewing each neuron activation as a feature. The learned decision tree can capture many different activation patterns in the concerned layer and can serve as a formal (logical) interpretation for that particular layer.  In contrast, we aim to find a dominant neural activation pattern (across layers) and further verify that it captures a large fraction of desired inputs from a specific class and meanwhile does not contain any adversarial input.
>
> [b]
> Sabour et al propose a novel approach to generating adversarial images. The proposed adversarial attack is formulated by solving a constrained optimization problem on neural representation at an intermediate layer. It is a very interesting related work in the sense that both this work and our work rely on neural activations but take the opposite angle.
> More broadly, we find the following interesting connection. The key connection between classic adversarial attack work and robustness verification work is the same focus on small L-k norm perturbations for a given input. Sabour et al’s work and our work share a very similar connection, except for that both concern neural activations, where perturbations on the input space can be quite large but stay close in the aspect of neural activations. For that reason, our work can be seen as a potential defense against the attack proposed in [b].

---

### Official Review · Reviewer_F47b · 2022-10-28

**Confidence:** 4
**Correctness:** 2
**Technical Novelty And Significance:** 2
**Empirical Novelty And Significance:** 2
**Recommendation:** 3

**Clarity, Quality, Novelty And Reproducibility:**

The work is Novel.

But I doubt the correctness of the claims and lack of proper experimental validation. Please see my detailed answer in the previous question.

QUESTION:

If you find a specification which all same class test examples follow, have you not solved ML?  If you have such a constraint, then just use that to make class prediction and your accuracy will be 100% ($- \delta$)?

**Strength And Weaknesses:**

## STRENGTHS
1. Finding alternative specifications is a good direction to explore

## WEAKNESSES

1. Writing needs to be improved
- Not put into context properly: Doing adversarial verification is not necessarily an *overfitted* specification, as you call it. There are many applications where we also want to verify against adversarial examples. I don't think you need to belittle previous work. Your work is complementary, and that is how you should write the paper.
- I don't agree with the entire 'data as specification' vs 'neural representation as specification' angle.  Because if one looks at equations in Sec3.2, you have the same formulation, but an extra constraint/specification. So you can write we add a new specification, instead of saying one vs the other.
- Many equations are not numbered. How should I refer to them?

2. Strong and probably incorrect assumptions
- What is the guarantee that test examples will definitely follow same NAP?
- What is the guarantee that adversarial examples will not follow NAP
- Isn't such an assumption against the whole point of verification?

3. Weak experimental validation
- One main limitation of adv verification methods is their inability to scale beyond small epsilons. This is especially critical for your work because for going upto the test inputs, you would require large epsilon. Current methods don't allow that. So your method might not work on bigger datasets at all. Whereas, adv verification will still work on big datasets with small epsilon.
- The experiments are only conducted on the MNIST dataset. I feel that MNIST is too easy and that is why your assumptions hold there. Since your assumptions are so strong, can you validate them on a bigger dataset?

**Summary Of The Paper:**

###  Problem
The paper tackles the problem of neural network verification, which aims to give guarantees of whether a trained NN follows certain specifications.

###  Proposed method
The authors claim that the current adversarial robustness specification used by the verification community is improper. Their claim is that this specification does not allow verification of test set inputs as they lie outside the verified region.

Authors propose neural representation as specification. Essentially, they propose to use neural activation patterns as specifications.

### Experimental Validation
Experiments are conducted on the MNIST dataset, where it shows that with the proposed specification, the certified region can go well beyond adversarial examples and even verify test set inputs.

### My one line understanding
Authors claim to have found a condition which:
1. adversarial examples don't follow, so verification won't stop at them and will be able to go beyond
2. test examples from same class follow, so test set examples can be verified

**Summary Of The Review:**

It is important to look for alternate specification.

But the assumptions made in this work feel incorrect and are not properly validated, so I cannot draw a conclusion from the current version.

---

> ### Author Response · Authors · 2022-11-12
> **Clarifications on the Issues. Results for CIFAR10 with CNNs are added.**
>
> Thank you for the careful review and insightful feedback. We hope to clarify some of the points brought up in our responses below. This review points to several edits which we will certainly include in a revision.
> > Not put into context properly… Your work is complementary, and that is how you should write the paper.
>
> We argue for an additional new specification, our intention is never to belittle prior work. We will update the text to reflect this.
> > Many equations are not numbered. How should I refer to them?
>
> We apologize for this. We will revise the paper to fix this issue.
> > What is the guarantee that test examples will definitely follow the same NAP? What is the guarantee that adversarial examples will not follow NAP? Isn't such an assumption against the whole point of verification?
>
> We do not claim that all test examples must follow the same NAP. Rather, we claim that if inputs follow a NAP P^L, they all must be classified by the network as the label L. There is no guarantee for inputs that do not follow a (dominant) NAP. We note that the dominant NAPs (Def. 3.1) are too strong. We need to relax the definition to delta-relaxed dominant NAPs (Def. 3.2) that are more forgiving. Given a dataset, we find a delta-relaxed dominant NAP candidate P for each class L, and then, use verification, to check that if an input follows P^L, it must be classified by the network as L. (i.e., there are no adversarial examples “covered” by P^L). NAPs do not have to always exist – this is what makes the work feasible. However, we show they do exist (for a reasonably large fraction of train/test dataset) in real NNs.
> > Your work can verify large epsilon, which existing methods cannot achieve. Would your approach be able to verify bigger datasets with large epsilon?
>
> In terms of scalability, our approach should be the same as (or sometimes slightly better than) existing methods. The reason is that we refine the search space of the original verification problem by introducing NAP constraints (which is one main contribution), and we use the out-of-the-box neural verification engine. The augmented specification enables existing verification techniques to verify a much larger regions of inputs, which is particularly interesting because many testing inputs beyond reach of existing methods can now be formally verified.
> We believe that our method is complementary to other adversarial verification techniques. The goal of the current paper is to argue for the specification and its importance / usefulness. We leave to future work to explore scalability, e.g., design new and specialized verification techniques by utilizing unique properties of NAP.
> > The experiments are only conducted on the MNIST dataset. I feel that MNIST is too easy and that is why your assumptions hold there. Since your assumptions are so strong, can you validate them on a bigger dataset?
>
> We have extended our experiments to CIFAR10 and Convolutional Net.
>
> Results for CIFAR10 (the biggest image dataset in VNNCOMP2021) with a convolution net used in the same competition. We extract all (input, epsilon) pairs that are known to be not robust from the competition and index them x_0 to x_6. Without our specification (i.e. augmented NAP), it is impossible to verify an eps larger than 0.012
> | **Epsilon**    | 0.012 |      |     | 0.024 |      |     | 0.12 |      |     |
> |--------------|-------|------|-----|-------|------|-----|------|------|-----|
> | **Delta**  | 0.01  | 0.05 | 0.1 | 0.01  | 0.05 | 0.1 | 0.01 | 0.05 | 0.1 |
> | O(x_0)=8 | **Y**    | **Y**   | **Y**  | N     | T/o  | **Y**  | T/o  | **Y**   | **Y**  |
> | O(x_1)=6 | T/o   | N    | **Y**  | N     | N    | **Y**  | N    | N    | **Y**  |
> | O(x_2)=0 | **Y**    | **Y**   | **Y**  | **Y**    | **Y**   | **Y**  | N    | N    | N   |
> | O(x_3)=1 | N     | N    | N   | N     | N    | N   | N    | N    | N   |
> | O(x_4)=9 | N     | **Y**   | **Y**  | N     | N    | N   | N    | N    | N   |
> | O(x_5)=7 | **Y**   | **Y**   | **Y**  | N     | T/o  | **Y**  | N    | **Y**   | Crash  |
> | O(x_6)=3 | **Y**    | **Y**   | **Y**  | **Y**    | **Y**   | **Y**  | N    | N    | N   |
>
> > If you find a specification which all same class test examples follow, have you not solved ML? If you have such a constraint, then just use that to make class predictions and your accuracy will be 100% (−δ)?
>
> As discussed in Section 4.2, the accuracy comes with the tradeoff in recall. We found a pattern P such that if P(x) = True, then x must be classified as 1. However, we make no claim about x when P(x) = False. For example, in MNIST, there are 176 (of of 1135) images of digit 1 that do not follow P.

---

### Decision · Program_Chairs · 2023-01-20

**Decision:**

Reject

**Justification For Why Not Higher Score:**

More evidence about scalability, or alternatively other benefits of NAPs compared to traditional specifications.

**Justification For Why Not Lower Score:**

N/A

**Metareview: Summary, Strengths And Weaknesses:**

This paper introduces a new type of specification for verifying neural networks, based on activation patterns of neural networks (e.g., a set of units that are assumed to be greater than 0, and a set less than 0). The idea is to verify that certain properties of a network hold, conditional on the activation pattern holding.

There was significant debate among the reviewers about this paper, which led to an active discussion.

There was general agreement that researching alternative specifications was a good direction to explore, and several reviewers judged that the overall direction of using activation patterns was a potentially promising approach.

Several reviewers raised concerns about the overall framing of approach (e.g., that standard adversarial specifications may be “overfitted” etc.). It is good that the authors agreed to tone down this language in their response.

The main negative concern was about the scalability. Certainly having experiments only on MNIST, as the original draft did, would not be sufficient for publications. In the response, the authors add additional experiments on CIFAR with a different architecture. This strengthens the work considerably.

However, there were still concerns raised in the discussion about scalability. Even with these new results, the CIFAR architecture is within the range of networks that can be verified using traditional specifications. So the question was raised about whether NAPs provide a scalability benefit. This seems particularly important to me because NAPS come with a significant decrease in interpretability, so it is much harder for an engineer to check that the specification is correct. Increasing the evidence to support the benefits of NAPS as a specification, while being upfront about their drawbacks, would significantly improve the paper.